

# Evaluation of a combined drought indicator and its predictive potential for agricultural droughts in Southern Spain

María del Pilar Jiménez-Donaire[1], Ana Tarquis[2,3], Juan Vicente Giráldez[1,4]

[1] Dept. of Agronomy, University of Córdoba, Córdoba, 14071, Spain
[2] CEIGRAM, Universidad Politécnica de Madrid, Madrid, 28040, Spain
[3] Grupo de Sistemas Complejos, Universidad Politécnica de Madrid, Madrid, 28040, Spain
[4] Institute for Sustainable Agriculture, CSIC, Córdoba, 14071, Spain

Correspondence to: María del Pilar Jiménez-Donaire (p.jimenez.donaire@gmail.com)

**Abstract.** Drought prediction is critical, especially where rainfall regime is irregular, such as in Mediterranean countries. A

new combined drought indicator (CDI) is proposed that integrates rainfall, soil moisture and vegetation dynamics. Standardized precipitation index (SPI) is used for evaluating rainfall trends. A bucket-type soil moisture model is used to keep track of soil moisture and calculate anomalies, and, finally, satellite-based NDVI data is used for monitoring vegetation response. The proposed CDI has four levels, in increasing amount of severity: watch, alert, warning type I and II.

This CDI was then applied over the period 2003-2013 to five study sites, representative for the main grain-growing areas of

SW Spain. The performance of the CDI levels was assessed by comparison against observed data on crop damage. Observations show a good match between crop damage and CDI. Important crop drought events in 2004-2005 and 2011-2012, marked by crop damage between 70 and 95% of the total insured area, were correctly predicted by the proposed CDI in all five areas.

## 1 Introduction

Drought is a recurrent phenomenon on the Earth surface triggered by lack of water, or an extended imbalance between the supply and demand in the precise expression of Hobbins et al. (2016), which may be appreciated under different forms (e.g. Wilhite 2000). Drought is one of the most important natural disasters that threatens our society. In spite of its relevance there is not a proper definition of drought, considering the delimitation of the times of start and end, Tannehill (1974) called drought the creeping phenomenon for this reason, and of the spatial extent of its effects.

Wilhite and Glantz (1985) distinguished four main types of droughts according to how the effects were noticed: (i) meteorological due to the scarcity of rainfall; (ii) hydrological detected by low streamflow; (iii) agricultural when soil water is not sufficient to maintain a crop; and (iv) socioeconomic whose influence affect the normal manifestations of the society.

Drought occurs worldwide but it is especially frequent in the Mediterranean region. In a recent analysis of a tree-ring based reconstruction of the summer season Palmer Drought Severity Index (PDSI) (e,g, Keyantash and Dracup 2002), denominated

Old World Drought Atlas (Cook et al. 2015), for the 1100-2012 period, Cook et al. (2016) have detected the gravity of the



recent events in the area, apparently induced by anthropogenic activity. Combining two drought indices, one meteorological, the Standardized Precipitation Index (SPI), for the water supply, and the other a hydrological index, the Standardized Precipitation-Evapotranspiration Index (SPEI), for the water loss tendency, Stagge et al. (2017) observed for the European continent in the period 1958-2014, that droughts were mainly driven by a temperature increase with the inherent increase of

the evapotranspiration rate, while rainfall did not change appreciably. In the southwestern United States Ting et al. (2018) found that under a $CO_2$ warming scenario, earlier spring drying was mainly due to a decreased mean moisture convergence. Frequently the drought occurs suddenly what is called flash drought driven by high temperature or by severe water deficits (e.g. Wang and Yuan 2018). Under the influence of global warming a hypothesis has been formulated according to which dry regions tend to become drier while wet regions tend to become wetter, the DDWW paradigm. Nevertheless, Yang et al.

(2019) have observed that at the global scale the paradigm is mainly confirmed in precipitation-driven drought, when the plant and soil conditions are not considered.

One additional problem of the drought is that it can progress towards other regions as Herrera-Estrada et al. (2017) have discovered in their Lagrangian analysis in several Earth regions. Andreadis et al. (2005) have elaborated severity-area duration maps modifying an earlier proposal of Dalezios et al. (2000) of severity-duration-frequency maps. Therefore,

drought is a present-day risk at least for a part of our society,

The characterization of the drought depends on the perspective of the user. The meteorological drought is possibly the simplest type to evaluate since it reduces to the consideration of the rainfall. The two main meteorological drought indices are the mentioned PDSI and SPI. The hydrological drought requires the conversion of rainfall in runoff what can be made with the help of a hydrological model. The SPEI is one widely used hydrological drought index. Nevertheless, Van Loon and

Van Lanen (2012) have explored in depth the definition of hydrological drought starting from the time perspective of the phenomenon, their Figure 1, and distinguishing several types in term of the sequence rain to snow, wet to dry, cold snow, warm snow seasons and what they denominated classical rain deficit. The use of a simple hydrological model and the establishment of some threshold values allow Van Loon and Van Lanen (2012) to determine the drought occurrence in several regions with distinct climate types. The severity of the drought is a function of the available water storage units as

Van Loon and Laaha (2015) explained in the survey of an Australian dataset. Hobbins et al. (2016) have modified the SPEI index by representing the potential evapotranspiration, the atmospheric evaporative demand, in a proper physical basis, rather than on the air temperature as a proxy of it. Their drought index, the evaporative demand drought index (EDDI) is a useful indicator of the drought extent as shown by McEvoy et al. (2016) in the conterminous US. The estimation of the agricultural drought index is rather similar to that of the hydrological drought index, with the additional complexity of the

crop behavior. Several models have been proposed for the estimation of the agricultural drought index. As Perrin et al. (2001) warned, and later Orth et al. (2015) confirmed, the models to describe soil water evolution for this purpose must be very simple, limited to a soil water balance. Hunt et al. (2009), Khare et al. (2013), and Sohrabi et al. (2015) proposed reasonable models of soil water balance differing only in the characterization of the rainfall infiltration, to discard the generation of excess rain, deep percolation, and actual evapotranspiration rate.



The different drought indices represent distinct aspects of drought. Therefore, to gain a wider perspective Kao and Govindaraju (2010) introduced the use of copulas in a new drought indicator denominated the joint deficit index (JDI) based on the SPI for both precipitation and streamflow. Hao and AghaKouchack (2013) formulated another copula, the multivariate standardized drought index (MSDI), consisting of the SPI and of a standardized soil moisture index (SSI). This

index was very useful to detect the onset and the duration of the drought. Alternatively, Zarch et al. (2015) used two separate indices to assess droughts, the SPI and the reconnaissance drought index (RDI). A different approach was suggested by Hao et al. (2016) with a categorical drought prediction, the U.S. Drought Monitor (USDM), which proved to be very adequate for early warning. Azmi et al. (2016) developed a data fusion-based drought index, grouping different indices with a clustering method.

The impact of drought on vegetation can be appreciated through several indices. Kogan (1965) proposed a vegetation condition index (VCI) based on the normalized difference vegetation index (NDVI), which is a good indicator of the state of the vegetation combining the radiance of the visible and infrared wavelengths to assess the effects of the drought. Some other indices have been suggested, since NDVI is sometimes influenced by other environmental factors (Quiring and Ganesh 2010). The normalized difference water index (NDWI) was introduced by Gao (1996), using radiances in a range of greater

wavelengths than NDVI is less affected than the latter by atmospheric conditions, and it is also more sensitive to drought than other indices (e.g. Gulágsi and Kovács 2015). The Joint Research Center of the European Union uses the fraction of absorbed photosynthetically active radiation (fAPAR) generated from the signals acquired by an especial sensor.

The above-mentioned methods can be used to evaluate the impact of drought on agricultural productivity of the world regions as Sepulcre-Cantó et al. (2012) have shown for Europe. For directing local policy actions and mitigation actions,

such as farm-scale insurance schemes, smaller spatial scales than those used by Sepulcre-Cantó et al. (2012) are required.

The main objective of this work is to assess agricultural drought by means of a combined drought indicator (CDI), based on SPI and anomalies in soil moisture and NDVI. This new CDI is then related to data on crop damage in rainfed wheat producing regions in Southern Spain, at the level of agricultural provinces, which corresponds to the highest detail of available yield data.

## 2 Materials and methods

### 2.1 Study area

This study was made in Andalusia, South Spain, during the 10-year period between 2003-2013.

Andalusia has a Mediterranean climate dry and hot summers, (Köppen-Geiger climate Csa, Peel et al. 2007). Since the main source of the water is the rain caused for the Western and Southwestern winds carrying the moist air from the Atlantic Ocean

the distribution of the precipitation is conditioned by the orography of the region, with a main decreasing gradient from the west to the east.



The effect of drought on agricultural production was evaluated in five representative areas. In each of these, 4 representative locations were selected in a two-step procedure. First, the distribution of the land use class "non irrigated arable land" within the study area was analyzed, shown in figure 1. This land use distribution is derived from the regional land use map (SIOSE - Sistema de Información de Ocupación del Suelo de España aplicado a Andalucía, 2005),equivalent to the European

CORINE database, at a scale of 1:10.000. This class occupies 20 %, 886.250 ha, of the total utilised agricultural area within Andalucia, 4.402.760 ha (Censo Agrario, 2009, INE). Although the class of "non-irrigated arable land" also includes other non-cereal crops, in our study area wheat is by far the dominant crop. Second, five agricultural districts within Andalusia were selected where this crop is dominant: Campiña de Cádiz (Cádiz), Campiña Baja (Córdoba), Pedroches (Córdoba), Norte o Antequera (Málaga) y La Campiña (Sevilla). In each of these districts, four representative point locations were

selected, yielding a total of 20 point locations. These point locations correspond to 250 x 250 m pixels, equivalent to the resolution of the NDVI imagery (see point 2.3). The selection of these pixels was carefully made and subject to a visual case-by-case analysis in order to exclude anomalies and assure a homogeneous land use in the following remote sensing analysis. Each of the 20 point locations had to fulfil the following conditions, that were checked manually using aerial ortophoto imagery from 2004 to 2013:

(i) presence of a homogeneous land use of rainfed wheat within each pixel (no other land uses present within the pixel).

(ii) absence of external landscape elements, such as ponds, roads, channels, houses or natural vegetation patches that could distort the NDVI signal.

(iii) continuous wheat cultivation during the study period (no fallow).

**2.2 Standardized Precipitation Index (SPI)**

The SPI was calculated over 1, 3  and 6 months periods using precipitation series between 42 and 69 years,  namely SPI-1, SPI-3 and SPI-6.

SPI-1 is theoretically best related to meteorological drought, together with short-term soil moisture stress, especially in periods where crop growth is sensitive to these (Guttman, 1999). SPI-3 has been shown to reflect short to medium seasonal precipitation trends (Guttman, 1999). Bussay et al., (1999) and Szalai and Szinell (2000) evaluated the relation between SPI

and agricultural drought through soil moisture and found SPI-2 and SPI-3 to yield the best results. Other authors (Ji and Peters, 2003; Rossi and Niemeyer, 2012) have shown a high correlation between SPI-3 and vegetation response and therefore concluded this index to be the better suited for evaluating agricultural drought. SPI-6 is then better suited to identify longer-term or seasonal drought trends.

The code SPI-SL 6, developed by the National Drought Mitigation Center, University of Nebraska-Lincoln, was used to

calculate SPI. Details of the method can be found in McKee et al. (1993) and Lloyd-Hughes and Saunders (2002).

The same classification used by McKee et al. (1993) was used (Table 1) and a threshold value to define a drought of SPI > - 1,00 was used, following Cancelliere (2004).





SPI values were calculated for each of the five agricultural regions selected: Campiña de Cádiz (Cádiz), Campiña Baja (Córdoba), Pedroches (Córdoba), Norte o Antequera (Málaga) y La Campiña (Sevilla). Within each region, the climate series selected was the one the station that had the longest available series within that particular region.

### 2.3 Soil Moisture Anomaly Index (SMAI)

Soil Moisture Anomaly Index (SMAI) values were calculated for each of the five selected agricultural regions, similar to the SPI. To evaluate soil moisture dynamics, the simple water balance model of Brocca et al. (2008) was used. In this model the water depth in the soil profile, W, evolves with time, t, following the contribution of the infiltration of the rain, f, and the extraction of the evapotranspiration, e, and of the deep percolation or of the surface and subsurface runoff, g. The balance was computed at the daily time scale following Eq. (1)

$$\frac{dW(t)}{dt} = f - e - g \tag{1}$$

The infiltration depth is estimated from the rain depth, p, the wetness or relative soil water content, normalized by the maximum value, $W_{max}$, $\omega = W/W_{max}$. and a parameter m, with the empirical approximation proposed by

Georgakakos (1986), Eq. (2):

$$f = p(1 - \omega^m) \tag{2}$$

The deep percolation or runoff loss is estimated by a simple potential function with the saturated hydraulic conductivity, $k_s$, and $\lambda$ the index of pore size distribution of Brooks and Corey (1966), Eq. (3):

$$g = k_s \omega^{3+2/\lambda} \tag{3}$$

Finally, the daily evapotranspiration rate is estimated as the FAO-Penman Monteith (Allen et al., 1998) potential rate, $e_0$, modified by the wetness, Eq. (4):

$$e = \omega e_0 \tag{4}$$

The parameter values adopted here are shown in Table 2.


The Soil Moisture Anomaly Index (SMAI) is then given by Eq. (5):

$$SMAI = \frac{W - \overline{W}}{\sigma_W} \tag{5}$$

where $\overline{W}$ is the long-term average soil moisture and $\sigma_W$ its standard deviation.





### 2.4 NDVI anomaly index

Different studies of agricultural drought have used satellite-based vegetation indices, as their main advantage is their spatial and temporal resolution. NDVI values represent the plant chlorophyll content, which is why they are highly suitable for identification of agricultural drought. Limitations in its use are related to the fact that NDVI can reflect non-drought related

stress conditions, such as plant disease, and that soil properties can induce a bias in its response. Therefore, it is important to use NDVI-based drought evaluation in combination with other indices based on precipitation or soil water, as is the case here. NDVI anomalies are evaluated on a monthly basis, but only taken into account from November to April, which is the normal growing season for winter rainfed cereal in Andalucia. Only during this period it can be expected that NDVI and its anomalies inform about rain fed cereal growth.

Thanks to its spatial continuity, NDVI trends could be analyzed for 20 different points, as four points or pixels were analyzed within each of the five selected agricultural regions. This analysis yielded a total of 20 spatially different NDVI anomaly indices. The NDVI anomaly index was calculated as, Eq. (6):

$$\text{NDVI anomaly index} = \frac{\text{NDVI}_i - \overline{\text{NDVI}}}{\sigma_{\text{NDVI}}} \tag{6}$$

Where $\text{NDVI}_i$, $\overline{\text{NDVI}}$ and $\sigma_{\text{NDVI}}$ are respectively its value at a particular moment in time, its long-term mean value, and its standard deviation. NDVI data was derived from Terra MODIS that collects imagery for each point on Earth every 1-2 days. Based on this data, a monthly average was calculated and used for $\text{NDVI}_i$ (Consejería de Agricultura, Pesca y MedioAmbiente. Junta de Andalucía).

### 2.5 Combined drought indicator (CDI)

The main idea behind the combined drought indicator (CDI) for identifying agricultural drought is an idealized cause-effect relation between water deficit and yield. A precipitation deficit leads initially to a soil water deficit, which if prolonged over time will result in crop water stress, and reflect in the observed NDVI, which finally results in a reduction of cereal yields.

In its simplest form, this CDI would allow to identify which phase of the cause-effect relation the agricultural system has

attained in the event of a drought. This indicator would then allow to establish a series of drought warnings depending on the phase. The CDI should be seen as a first step to designing such a warning system.

This study proposes a CDI that combines three combines, as mentioned before:

- SPI-3 to identify the first level of precipitation deficit
- SMAI to identify anomalies in the soil moisture

- NDVI anomalies to characterize the subsequent effect of soil water stress on crops.

The proposed warning levels of the proposed CDI are given in Table 3. It can be expected that these help policy makers to prepare and take action in the case of droughts.



The CDI uses three different levels, the first two levels of attention and alarm indicate that a drought could be imminent. The highest level of the CDI is the warning. The two types of warning take into account those cases where a meteorological drought results in a rapid yield decrease. The type I warning can occur even without a previous anomaly in soil moisture values. This could be related to intense droughts occurring during sensitive phenological phases of the crop. Therefore, a

type I warning depends only on two indicators, SPI-3 and NDVI. The type II warning is based on all three indicators that compose the CDI (SP-3, SMAI and NDVI) so that these give a more solid evidence for the existence of an agricultural drought.

### 2.6 Insurance data

The insurance area and the affected area by drought per agricultural campaign, for rainfed cereal, were given by Agroseguro.
This data was disaggregated for each area, of the five under study, and each agricultural campaign, from 2002/03 till 2011/12. Crop intensity damage is expressed as the percentage of surface area that was filed for damage with respect to the total insured area, and are available at the agricultural region scale. Crop damage close to 100% indicates important losses during that year.

## 3 Results

### 3.1 SPI

The SPI values calculated over a three-month period (SPI-3) reflect moisture conditions over short-medium term, and offers an estimate of the seasonal precipitation, that is useful for agricultural purposes. In our area, SPI-3 values at the end of April reflect the precipitation trends during the plant's reproduction stage and the grain development. SPI-3 at the end of December reflects moisture conditions at the start of the growing season.

Figure 2 shows the trends in SPI-3 for all five selected agricultural regions. The trends are similar in all regions, with SPI-3 values moving periodically around the long-term mean or 0 value. In the driest years, one can observe the highest negative peaks. For example, during the agricultural year 2004-2005 which was very dry, negative values up to -2,50 can be observed for Campiña de Cádiz, indicating the severity of the drought. Another dry year was 2011-2012, where negative values of -2,12 can be observed during the month of February in La Campiña. So clearly, the two main dry periods are correctly
identified by the trends in SPI. However, this drought indicator also marks different other periods as critical that were not markedly dry. In 2008-2009 all regions are marked by critical SPI levels, albeit for short periods of time and mainly towards the summer or end of the agricultural year. Even in 2012-2013 critical drought periods are flagged in four out of five regions.

### 3.2 SMAI

Figure 3 shows the variation of SMAI over the studied period and for each of the five studied agricultural regions. The main
two dry periods of 2004-2005 and 2011-2012 are not consistently apparent. Generally, only two regions at the time dip



below the -1 mark and are indicated in red: (a) Campiña de Cádiz and (d) Norte/Antequera for 2004-2005 and (a) Campiña de Cádiz and (b) Campiña Baja for 2011-2012. The year 2007-2008 seems to be marked by drier soil water contents compared to the long term mean, as critical levels are reached for four out of five agricultural regions.

### 3.3 NDVIA

Figure 4 shows a map indicating the spatial and temporal variability in NDVI values over Andalusia. Figure 4a indicates NDVI in April, right in the growing season while Figure 4b shows the same area after the cereal has been harvested. Red colors indicate low values of NDVI while green colors indicate maxima between 0,96 in April and 0,92 in June. When comparing to the distribution of the main cereal growing regions in the area in Figure 1, it can be seen that these areas present the most important variation between the two images, with high values in April and low red values in June.

Figure 5 shows the monthly variation of the NDVI anomaly for the four selected pixels within the Campiña agricultural region. The pixels in the other four agricultural regions are not shown, but their trend is similar. There is of course an important spatial variability within the area, so that some differences appear between the four study locations. This can be attributed to different planting dates, different varieties or different soil properties between the locations. Over the study period however, the same general temporal trends appear. Important negative deviations from the mean indicate periods of

high plant stress. Values of NDVI anomaly below -1 are marked in red. Its evolution is similar to the evolution of SPI-3 and SMAI (Figures 2 and 3), although there is clearly a time lag effect. Plant stress generally only occurs after a precipitation and soil moisture deficit has occurred. Also, the temporal pattern is more erratic than in the case of SPI-3 and SMAI. However, the previously mentioned droughts of 2004-2005 and 2011-2012, can be identified as negative peaks in Figure 4. During other years, isolated red deviations appear, but these are not generalized among all four sites. The only exception is 2008-

2009 where a generalized NDVI anomaly appears in all, but it occurs early during the first months of the growing season so perhaps it can be attributed to a late seeding that year.

### 3.4 CDI

Figure 6 shows the evolution of CDI between 2003-13 and compares its levels against crop damage data derived from agricultural insurance information. This occurs twice during the studied period at a regionalized scale, indicating the effects

of a drought. The first time is during the agricultural year 2004-2005, with losses between 73-99% in the five studied agricultural regions. Also for the year 2011-2012 crop damage is high, between 71-92%. There is a third campaign, 2009-2010 with medium to high losses, between 44 and 89%. However, crop damage during this period is rather due to the effects of excessive precipitation, leading to water stagnation and erosion damage. This can be seen when comparing the annual precipitation values. For example, in the Cordoba agricultural region, with a mean long-term precipitation of 600 mm, the

values for 2004-2005, 2009-2010 and 2011-2012 are respectively 423, 1179 and 433 mm.

The CDI indicator accurately captures these two important drought periods. For the first area, Campiña de Cádiz (Figure 6a), a series of drought warning levels are issued early in the agricultural year 2004-2005, followed by a type I alert in January.





There is another type I and II alert in May-June. In other words, since the seeding and during the first months of the crop growth, there is a continued series of drought warnings or alerts. In that particular year, 90% of the insured area was reported as damaged. In 2005-2006, the CDI registers another warning indication, but it does not lead to a damage of the crop. In September 2005 there is a type II alert, but this month is outside the cereal growth period and when the crop is seeded, two

months afterwards, the situation has recovered. In May 2006 there is another warning issued, due to a precipitation and soil moisture deficit. However, the crop is already in a moment of its cycle where it is close to harvest and therefore not so affected. In 2009-2010, marked by a high crop damage of 89% of the total insured area, there is only one alert in November. As mentioned before, crop damage during this year was probably due to precipitation excess rather than drought. For the dry period of 2011-2012, the CDI accurately indicates this critical situation with a warning followed by type I and II alerts in the

period February-April.

For the Campiña Baja region also, (Figure 6b), the dry period of 2004-2005 is characterized by a continuous series of type I and II alerts from January to June, with two more alerts during summer, outside the cereal growing period. In this region, the damaged insured area is very great that year as well (95 %). In 2008-2009 there was a warning issued that did not provoked yield losses, as the damaged insured area is only 15%. This can be explained due to the fact that this situation did not occur

at a time where the crop was sensitive. In another dry year, 2011-2012, a series of warnings are issued, from January to March, followed by respectively a type II and I alert in April and May. These all occur at times where the crop is highly sensitive, so that important damage occurs. The damaged crop area amounts to 90%.

In the Pedroches region (Figure 6c), the two main dry periods are predicted well. The year 2004-2005 is marked by a series of type II alerts in January, February, March and May and a type I alert in June. This sequence of critical CDI levels, is

reflected in a damaged insured crop area of 73%. In 2005-2006, although there are two types of stress situations, warnings and type II alerts from November to February, there is no high damage rate. The damaged insured area is only 15 %. It is difficult to understand the underlying reasons for the good performance of the crop that year. For example, during the year 2008-2009, the incidents are clearly late in the year (May to July), a period where grain growth is not sensitive. The second dry period of 2011-12, is marked with a number of type II alerts issued from February to April, at a time when the cereal is

highly vulnerable. This is reflected in a 71% damaged insured area.

In the Comarca Norte/Antequera region (Figure 6d), the dry period of 2004-2005 is marked by several incidents early on, with a watch issued in November and a type II alert in January, being the period of cereal nascence and other sensitive periods. That year, the damaged insured area is 88%. In 2007-2008 there are two warnings and a type II alert, from December to February, but these do not lead to crop damage, as the damaged insured area is only 11%. Again, the reason

must be sought in these droughts occurring during a period when the cereal is not too sensitive. During the second main dry period of 2011-2012 a number of type I and II alerts are issued between February and April. These correspond to moments of the crop cycle that are highly sensitive, and damage insured areas are consequently high that year, amounting to 83%.

The last region, La Campiña (Figure 6e), shows a similar trend, with 2004-2005 being marked by an extremely high damaged insured area of 99%. The CDI worked well in predicting this, as there are multiple and continued alerts, from





January to June there is a continued type II alert, except March which is a type I alert. In 2009-2010 there is a watch in November, and the damaged area is 72%. However, as mentioned before, the absence of further drought watches during this year and the high total annual rainfall indicates that this damage is more probable to have been caused by excess precipitation. In the second main dry period of 2011-2012 the situation is worse, with a number of warnings from January to
March, a type II alert in April and again a type I alert in May. That year the damaged insured area is high, up to 90%.

**4 Discussion**

The results allow concluding that the performance of the newly proposed CDI is adequate (figure 6). The periods of high crop damage - between 70 and 95%- in the two important dry periods of 2004-2005 and 2011-2012 are accompanied by watches, warnings and alert levels of CDI in the five agricultural regions that were studied. This combined indicator has
several advantages over using a single indicator, as is evidenced by the trends in precipitation, soil moisture and vegetation alone. Soil moisture alone for example, does not pick up the two main dry periods, 2004-2005 and 2011-2012, in all areas. The soil moisture anomaly index only indicates drought in two out of five regions for each of these dry periods. Measurements of in-situ soil moisture could probably improve this. Krueger et al. (2017) for example showed how in-situ soil moisture measurements explained wildfire incidence much better than the widely used Keetch–Byram Drought Index
(KBDI). Like our SMAI, the KBDI is a drought index calculated at a daily scale, but it only takes into account daily temperature and precipitation to calculate soil moisture. While our SMAI uses a more advanced soil water balance algorithm (e.g. using variable infiltration rate and refining the estimation of the actual evapotranspiration rate from the potential rate computed by the FAO-Penman Monteith equation), it is clear that future studies should focus on site-specific calibration of soil moisture dynamics against field data or by observations from remote sensing. Martínez-Fernández et al.(2015)
successfully applied in-situ soil moisture measurements to predict agricultural droughts in Northern Spain. Other studies, like Kędzior and Zawadzki (2017) have used SMOS-derived soil moisture anomalies. They concluded that these were suitable to calculate agricultural drought risk in the Vistula river catchment. Another possibility for improvement of drought prediction based on soil moisture is to combine different models. Cammalleri et al. (2016)used ensemble means from three different models, LISFOOD, CLM and TESSEL, and were able to increase the correlation with observations and reduce the
number of false drought alarms.

In any case, our results corroborate previous studies using combined indicators that also concluded that these yielded good results for agricultural drought prediction. Sepulcre-Canto et al. (2012) for example use a similar CDI, based on SPI, soil moisture and Photosynthetically Active Radiation (fAPAR). They evaluate this indicator at the continental scale and assess its performance against annual cereal yield at the regional level. They conclude their indicator to be successful in predicting
drought periods and lower yields. While our indicator is similar in conception, there are notable differences with the CDI proposed in this study, firstly in the way soil moisture anomalies are calculated, and secondly by using NDVI instead of fAPAR. Gouveia et al. (2009) comparing a soil water index against NDVI response in Portugal, found a good correlation





between NDVI and soil water content in different land use conditions. They concluded that NDVI values of arable land was more sensitive to drought compared to forests, which suggests that NDVI is particularly well suited in this study of cereal-growing areas.

Future studies could focus on improving this combined indicator, for example by using other probability density functions
than the gamma function used for calculating the SPI. Sienz et al. (2012) obtained a better fit to precipitation data of several world region with the Weibull than with the gamma probability distribution function. Carrão et al. (2016) selected an empirical standardized soil moisture index, which was highly correlated ($r^2$=0.82) with the maize-soybean and wheat yields to their in three study sites in Argentina.

### 5 Conclusions

This study has presented a new combined drought index (CDI) to evaluate agricultural drought. This CDI uses a combination of anomalies in precipitation (SPI-3), soil moisture and NDVI. The alert results are classified in four levels ranging from watch, warning to alert (type I and II). The CDI dynamics have been assessed for a 10-year period between 2003-2013, characterized by two important drought periods (2004-2005 and 2011-2012), in the five main rainfed cereal growing regions of SW Spain. Comparison with yield data shows that both dry periods, characterized by a high crop damage extent between
70 and 95%, are correctly identified by different critical CDI levels in all five study regions. This demonstrates the potential of this CDI. Further research should focus on a better representation of soil moisture data, either by improving data input from in-situ measurements or by remote sensing, or by using model ensembles. Also phenological information could be used to improve the performance of this indicator.

### Acknowledgments

The senior author was supported by a CEIGRAM fellowship. The insurance data were gently supplied by Agroseguro.
Second author is grateful to the Comunidad de Madrid (Spain) and Structural Funds 2014-2020 (ERDF and ESF) for the financial support (project AGRISOST-CM S2018/BAA-4330) and EU project 821964 – BEACON.

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

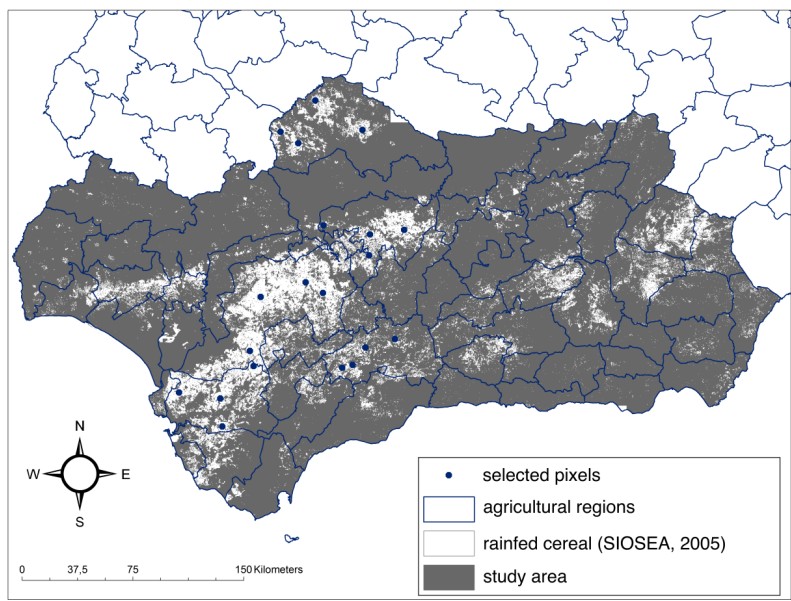

**Figure 1. Location of the study area (grey) and selected representative points (blue dots) within the areas cultivated with cereal (white).**





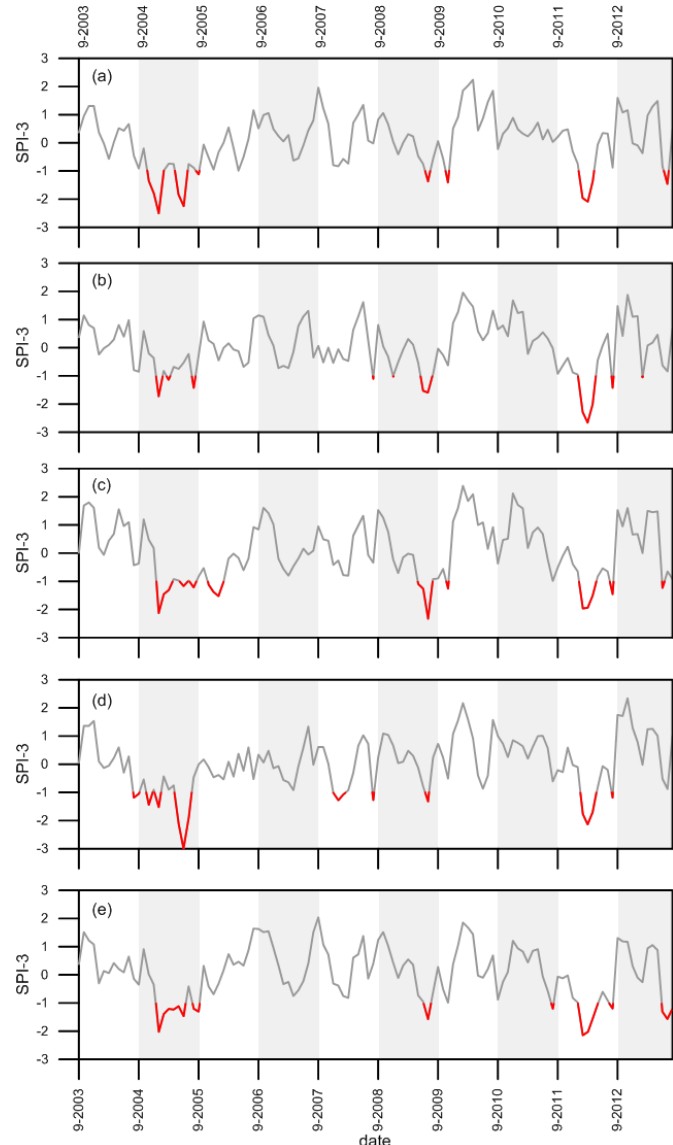

**Figure 2. Variation of the standardized precipitation index over 3 months (SPI-3) during the studied period (2003-2013) in the five selected agricultural regions: (a) Campiña de Cádiz; (b) Campiña Baja; (c) Pedroches; (d) Norte/Antequera; (e) la Campiña. Red lines indicate values below the defined threshold.**





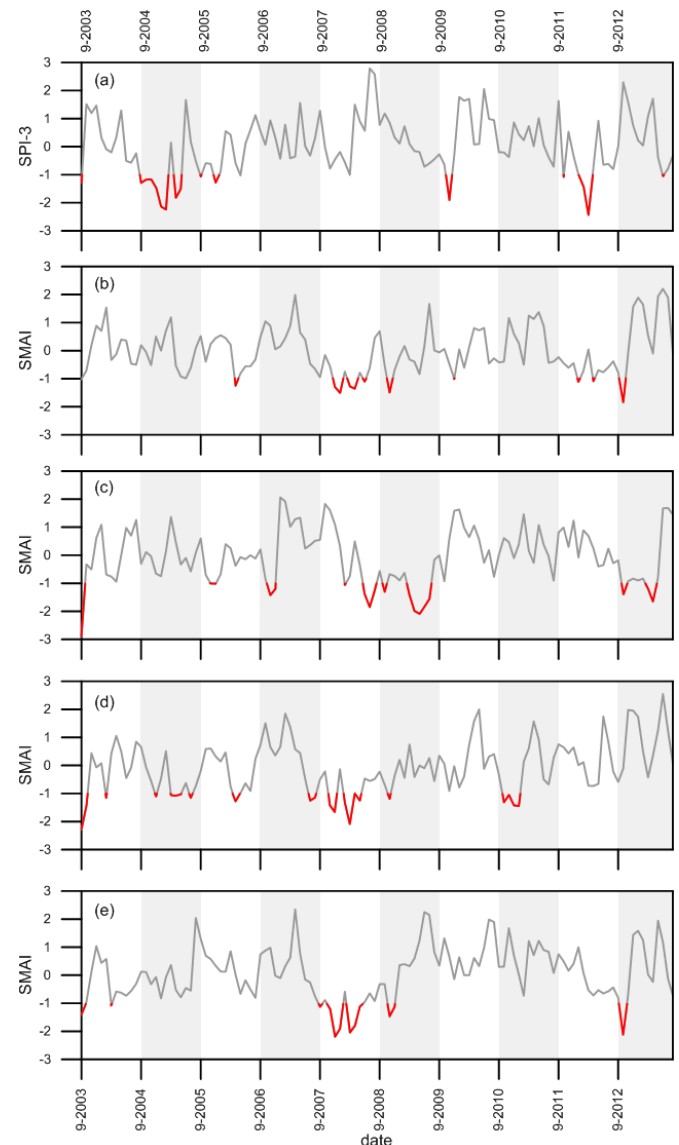

**Figure 3. Variation of the soil moisture anomaly index (SMAI) during the studied period (2003-2013) in the five selected agricultural regions: (a) Campiña de Cádiz; (b) Campiña Baja; (c) Pedroches; (d) Norte/Antequera; (e) la Campiña. Red lines indicate values below the defined threshold of -1.**





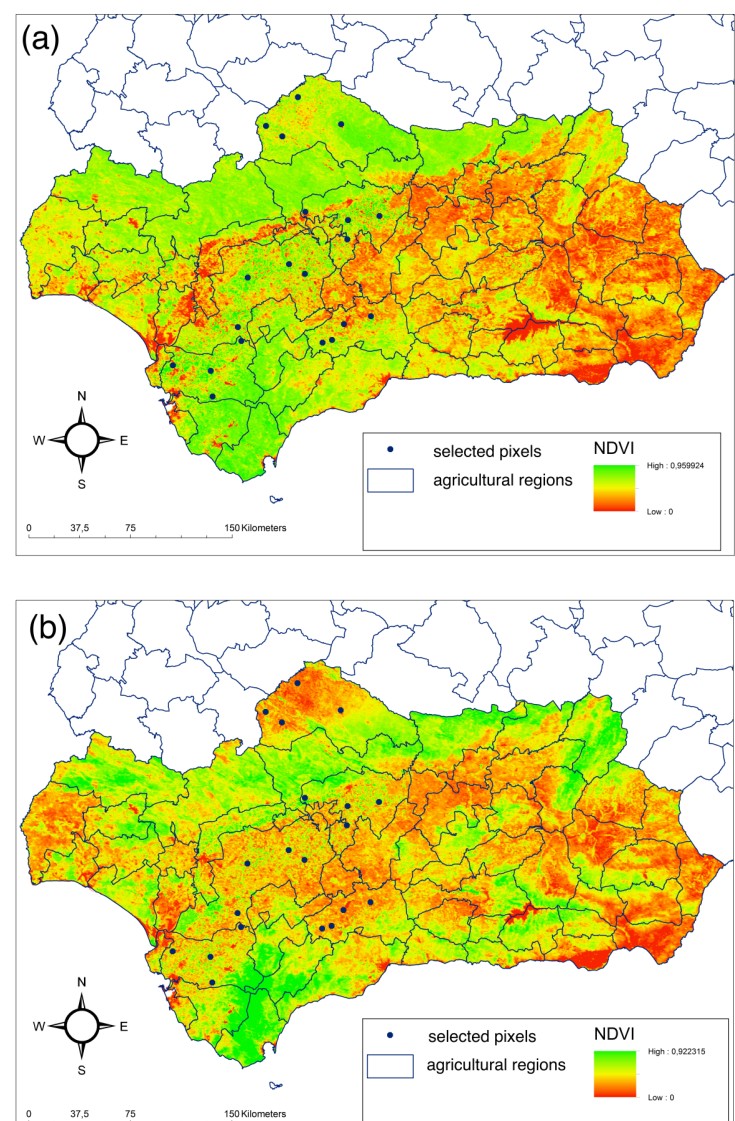

**Figure 4.** NDVI values over Andalusia in (a) April 2004 and (b) June 2004. Important changes from green to red are observed in the main grain-growing areas, while areas with natural forests and shrubs remain green. Blue dots show the four representative
5  pixels that were selected within each of the five studied agricultural regions.





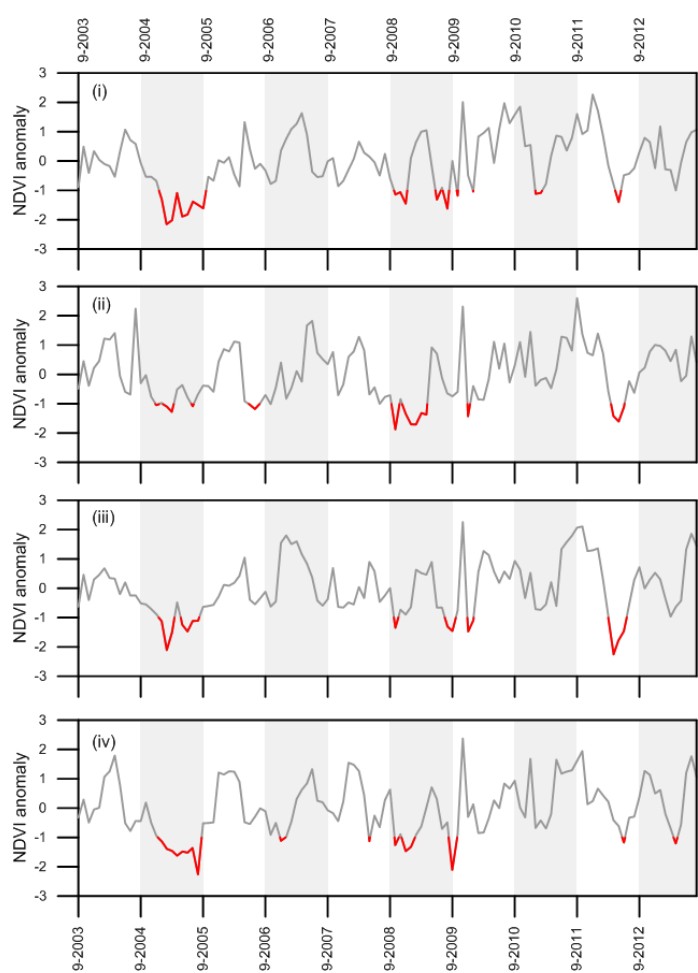

**Figure 5. Variation of the monthly NDVI anomaly for the four selected locations within the *Campiña* region over the study period. Red lines indicate values below the defined threshold of -1.**

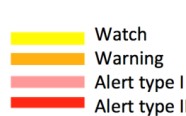





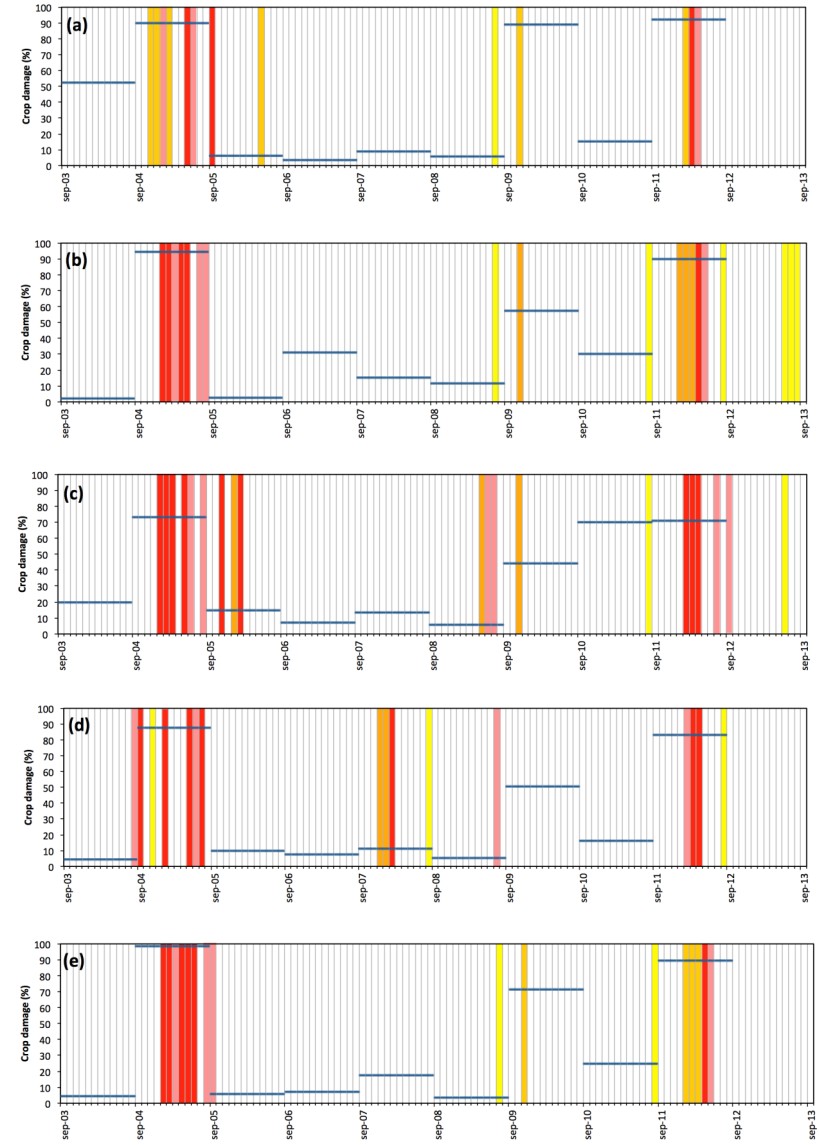

**Figure 6. Evolution of the Combined Drought Indicator (CDI) between 2003-2013 and comparison with agricultural crop damage intensity (blue lines) for the 5 agricultural regions studied: (a) Campiña de Cádiz; (b) Campiña Baja; (c) Pedroches; (d) Norte/Antequera; (e) la Campiña.**





| SPI | Category | Probability (%) |
|---|---|---|
| ≥ 2,00 | Extremely wet | 2.3 |
| 1,50 a 1,99 | Severely wet | 4.4 |
| 1,00 a 1,49 | Moderately wet | 9.2 |
| 0,00 a 0,99 | Mildly wet | 34.1 |
| 0,00 a -0,99 | Mild drought | 34.1 |
| -1,00 a -1,49 | Moderate drought | 9.2 |
| -1,50 a -1,99 | Severe drought | 4.4 |
| ≤ -2 | Extreme drought | 2.3 |

**Table 1. Classification of droughts according to SPI and their probability of occurrence following McKee et al. (1993)**

| Parameter | Value | Source |
|---|---|---|
| m (-) | 10 | mean value of the interval proposed by Brocca *et al.* (2008). |
| $W_{max}$ (mm) | 175 | as proposed by Vanderlinden (2001) calculated from the soil map of Andalucia |
| $K_s$ (mm day$^{-1}$) | 38.4 | Estimate of soil water properties by Rawls *et al.* (1998); representative value for loam clay according to USDA classification. |
| λ(-) | 0.15 | Derived from graphics of the parameter λ of Brooks and Corey (1966) as a function of soil texture, organic matter content and increase of soil porosity above the reference(Rawls *et al.*, 1983). |

5    **Table 2. Parameters for the water balance model used in this study.**

| Level | Definition | Characteristics | Situation | Actions |
|---|---|---|---|---|
| Watch | SPI-3 < -1 | Relevant precipitation deficit observed | Probability of Agricultural Drought occurring | ·Surveillance of the situation ·Prepare actions. |
| Warning | SPI-3 < -1 and SMAI < -1 | Relevant precipitation | Agricultural Drought expected | ·Put in place response strategies |





| | | deficit translates into an anomaly (deficit) in soil moisture | | for minimizing drought exposure |
|---|---|---|---|---|
| alert type I | SPI-3 < -1 and NDVI anomaly < -1 | Precipitation deficit is accompanied by an anomaly in vegetation condition: precipitation deficit leads to water stress in cereal | Agricultural Drought has started to affect yield negatively | ·Fortifying response strategies ·Careful follow-up of the situation |
| alert type II | SPI-3 < -1, SMAI< -1 and NDVI anomaly< -1 | Precipitation and soil moisture deficit are accompanied by anomalies in the vegetation condition: Water stress in cereal after precipitation and soil moisture deficit | Agricultural Drought has started to affect yield negatively | · Fortifying response strategies ·Careful follow-up of the situation |

**Table 3. Classification of the Combined Drought Indicator (CDI).**