# Peer review of "Evaluation of a combined drought indicator and its potential for agricultural drought prediction in Southern Spain"

_Natural Hazards and Earth System Sciences, 2019_

## Referee Comment (RC1) · Anonymous Referee #1 · 25 May 2019

This is an interesting work. The idea of combining soil, atmosphere, and vegetation status in one index reflects the developing holistic understanding of the soil-vegetation-atmosphere system. There are two issues that preclude the publication of the manuscript in its present form. A. Four important questions are not answered.

1. How did the authors arrive to the number of classes and boundaries shown in Table 3? 2. How did the authors evaluate the index? What was the objective way to do that? The Fig. 6 looks undoubtedly good, but no method of the index quantitative assessment is provided in the manuscript. 3. Does the proposed index work better than previously proposed indices? 4. What is the purpose of the index development?

Who and how will use it?

B. The English is unsatisfactory. Many statements are incomprehensible. Here are examples from the P. 1. L 21 "appreciated under different forms" What does this mean? L. 23 "proper definition" What does this mean? L. 24 "phenomenon for this reason, and of the spatial extent of its effects." What does this mean? L. 27 "influence affect the normal manifestations of the society" What does this mean? L 29 – 30 "denominated Old World Drought Atlas" What does this mean?

Terms are used that have not been defined.

Examples 6/25 "which phase" the phase was never defined SPI-3 to identify the first "level of precipitation deficit" What levels are you talking about. 6/27 "This study proposes a CDI that combines three combines, as mentioned before"

Some text pieces reflect simple negligence. Examples "representative value for loam clay according to USDA classification." Does not exist 4.3 NDVIA insurance data were gently supplied by

---

## Referee Comment (RC2) · Anonymous Referee #2 · 2 Jun 2019

This paper deals with the topic of defining a new combined drought indicator (CDI) capable to anticipate crop drought events. To do so, authors combined a meteorological indicator (SPI), a soil moisture indicator (SMAI) and a vegetation indicator (NDVIA). Authors established four levels of alerts with the corresponding actions and assessed this new indicator comparing monthly alerts with crop damage provided by the agricultural insurance. The research carried out in this paper is of interest, and I think it is adequate to NHESS journal. The manuscript is in general well-structured and the results that follows seems very reasonable to me. Correlation between the proposed CDI and crop damage is correctly presented. It seems to me that the manuscript could be published as long as the authors answer the following comments:

Specific comments: 1. Authors are using a different definition of the levels of damage crop in the abstract and in the results or conclusions. Are the levels "watch, alert, warning type I and II" (see abstract) or "watch, warning to alert (type I and II) (see conclusions)?. Regarding Table 3 it seems to be "watch, warning, alert type I and alert type II". 2. Could the authors extend the definition of SPI in "Methods"?. Some explanation of how SPI is calculated should be included to improve general understanding. 3. Could the authors explain how SMAI is calculated in the studied areas?. Did the authors obtain in-situ measurements?. How did you obtain the temporal evolution of SMAI in the studied areas? 4. Regarding your sentence: "Figure 3 shows the variation of SMAI over the studied period and for each of the five studied agricultural regions. The main two dry periods of 2004-2005 and 2011-2012 are not consistently apparent." Do the authors think that the information given by the calculated SMAI increase the accuracy of the drought prediction?. 5. NDVIA in four pixels have been calculated for every region. Could authors explain how these pixels have been combined to obtain the NDVIA per region?. Is simply the average of the four NDVIA values? 6. The proposed CDI seems to be a modification of Sepulcro 2012 indicator. I think some comparison with the latter, at least some advantages and drawbacks, should be included in the discussion. Is CDI the name of a family of combined indicators or is specifically the name of one indicator?. Perhaps, to avoid misunderstandings, the name of the new proposed CDI should be modified to distinguish it from the Sepulcro's CDI.

Technical comments: 1. Pag. 1 – line 21/22: Review format references in the text. An example: (e.g. Wilhite 2000). 2. Pag. 2 – line 21: I suppose you are referring to a fig. 1 of another article. Clarify this please. 3. Pag. 4 – line 9: Replace "o" by "or" and "y" by "and". 4. Pag. 4 – line 29: What is the meaning of SPI-SL 6? 5. Pag. 5 – line 2: Replace "o" by "or" and "y" by "and". 6. Pag. 6 – line 27: "This study proposes a CDI that combines three combines.." I suppose you want to say "three indices". 7. Pag. 7 – line 9: What is Agroseguro?. Explain please. 8. Pag. 8 – line 5: Indicate fig. 4 is an example of the year 2004. 9. Pag. 8 – line 23: Indicate fig. 6 shows a monthly evolution. 10. Pag 10 – line 27 – 29: Move to Introduction. Authors should explain this

Sepulcro 2012 indicator in the introduction. 11. Pag. 17 – Figure 3: In the first graph (3a) replace SPI-3 by SMAI.

---

## Referee Comment (RC3) · Anonymous Referee #3 · 7 Jun 2019

General comments:

This paper proposes a new combined drought indicator (CDI) integrating rainfall (SPI-3), soil moisture (SMAI) and vegetation dynamics (NDVI). It is shown that this indicator is useful to predict dry periods. Therefore the research carried out in this paper is of scientific and practical interest, and in my opinion it is adequate to NHESS journal. The manuscript is in general well structured and presented. The methodology employed and the obtained results are well exposed. It seems to me that the paper could be published as long as the authors answer the following minor concerns that arose from the review process that I made.

[Figure]

Specific comments:

I only have a couple of minor specific comments to be answered by the authors. 1. Did the authors made a comparison between the new combined drought indicator (CDI) that they propose and other combined drought indicators? Could the authors include in the paper some comments in this direction? 2. Did the authors apply this new combined drought indicator on geographic areas of different characteristics with respect to the characteristics of the areas in Southern Spain analyzed in the present study? Would it be possible that the evaluation of the CDI indicator should be different?

Minor comments:

Page 4, line 9: There are a couple of words not in English. Page 5, line 2: The same comment as I did before. Page 11, line 12: The classification of CDI is not clearly exposed in this section. Page 16, Figure 2: Perhaps it should be more clear (and homogeneous) substituting the "9" that appear at the beginning of the years, by "sep", as authors did in Figure 6. Furthermore, why the year 2013 is not printed in the figure? Also, in the description of this figure, I suppose that "La Campiña" should appear with capital letters, as in the rest of the document. Page 17, Figure 3: Same comments that I made for Figure 2. Page 19, Figure 5: Same comments that I made for Figure 2. Page 19, in the bottom: I suppose that the meaning of the colors that appear at the bottom of this page should appear inside Figure 6 of page 20. Page 20, Figure 6: In my opinion, this figure needs to be clearer with respect to the CDI indicator, including the meaning of the colors, for a better reader comprehension. Furthermore, it is not clear to me why the value of the agricultural crop damage intensity (blue line) is not printed for the last year. Pages 20-21, Table 3: Could the authors improve the quality of this table?

---

## Author Comment (AC1) · 15 Jul 2019

**This is an interesting work. The idea of combining soil, atmosphere, and vegetation status in one index reflects the developing holistic understanding of the soil- vegetation-atmosphere system.**

We thank referee 1 for this positive evaluation.

**There are two issues that preclude the publication of the manuscript in its present form.**

**A. Four important questions are not answered.**

**1. How did the authors arrive to the number of classes and boundaries shown in Table 3?**

This work is based on other similar indicators that have been published in the literature. The boundaries of SPI are very close to those originally proposed by McKe; those for NDVI are close to those defined for the vegetation index fAPAR by Sepulcre-Cantó et al. (2012); and those for the soil moisture categories were like those used by the NDVI.

**2. How did the authors evaluate the index? What was the objective way to do that? The Fig. 6 looks undoubtedly good, but no method of the index quantitative assessment is provided in the manuscript.**

We agree that future work should focus on improving the evaluation. At present, the limited data available only allows a qualitative evaluation. In other words, when we see some watch, warning or type of alert, we check that this corresponds to high levels of yield loss and insurance claims. While we realize that it would be better to do so quantitatively, we want to stress that even this type of qualitative evaluation has only rarely been undertaken in previous studies, so we think it is highly valuable.

**3. Does the proposed index work better than previously proposed indices?**

We have reported in the paper that our combined index works better than, for instance, SPI alone, which is an index that is now frequently used in drought management. See page 10, lines 9 - 11. We have not made a full evaluation against other combined indices, which is beyond the scope of this study. It would imply using other models (such as LISFLOOD which was the model used to calculate soil moisture anomalies for the CDI indicator proposed by Sepulcre-Cantó et al., 2012) and working on different spatial scales, so it would make the presentation of this indicator and its evaluation too complex. However, in a new study, this could be done and we therefore agree that it would be of interest to study it in future work.

**4. What is the purpose of the index development?**
**Who and how will use it?**

This is an interesting observation. We hope that this type of drought index could be employed by different users: (i) scientists; (ii) policy analysts and technicians working in drought management; and (iii) insurance companies.

We mentioned this in the introduction, page 1 lines 19-20 " For the management of local policy  and mitigation actions, such as farm-scale insurance schemes, smaller spatial scales than those used by Sepulcre-Cantó et al. (2012) are required."but we have added another phrase to make it clearer:, page 1 lines 24-25: " It is expected that this new CDI will be useful at the local policy level and for planning farm-scale insurance schemes."

**B. The English is unsatisfactory. Many statements are incomprehensible. Here are examples from the P. 1. L 21 "appreciated under different forms" What does this mean? L. 23 "proper definition" What does this mean? L. 24 "phenomenon for this reason, and of the spatial extent of its effects." What does this mean? L. 27 "influence affect the normal manifestations of the society" What does this mean? L 29 – 30 "denominated Old World Drought Atlas" What does this mean?**

We have double checked the manuscript and  have also sent it to a professional native English speaker for correction.

-P1.L21. This phrase has been replaced by "...*which may have economic, social and environmental impacts*"

-P.1.L24 has been rephrased *" Tannehill (1974) called drought "the creeping phenomenon", given the complexity of accurately delimiting its start time and end time, and of adequately demarcating the spatial extent of its effects.  "*

-P1 L27 this fourth type of drought has been rephrased as " *(iv) socioeconomic when it affects the normal functioning of society.* "

-P.1 L 29-30 this is the name given to the published results of this study, see http://drought.memphis.edu/OWDA/

We have deleted the reference to its name and focused on describing the results of this study.

**Terms are used that have not been defined.**
**Examples 6/25 "which phase" the phase was never defined SPI-3 to identify the first "level of precipitation deficit" What levels are you talking about. 6/27 "This study proposes a CDI that combines three combines, as mentioned before"**

With "which phase" we referred to the previous sentence, that explains the different phases: " A precipitation deficit leads initially to a soil water deficit, which, if prolonged over time, will result in crop water stress, and be reflected in the observed NDVI observed, which finally generates a reduction in cereal yields. "

The same goes for "first level of precipitation deficit", we refer to the first thing occurring during a drought, i.e.  the absence of precipitation (which then leads to a soil moisture deficit, etc...)

As this was obviously expressed in a confusing way, we have adapted this paragraph completely. It now reads as follows:

*"The main idea behind the combined drought indicator (CDI) for identifying agricultural drought is an idealized cause-effect relationship between water deficit and yield. There are different phases in this relation: a precipitation deficit (phase 1) leads initially to a soil water deficit (phase 2), which, if prolonged over time, will result in crop water stress, and be reflected in the NDVI observed (phase 3), which finally a reduction in cereal yields (phase 4).*

*In its simplest form, this CDI would allow to identify which phase of the cause-effect relation the agricultural system takes reached in the event of a drought. This indicator would then permit the establishment allow to establish of a series of drought warnings, depending on the phase. The CDI should be seen as a first step towards designing that warning system.*

*This study proposes a CDI that combines three indicator variables:*

- *SPI-3 to identify the first level of precipitation deficit (phase 1)*
- *SMAI to identify anomalies in the soil moisture (phase 2)*
- *NDVI anomalies to characterize the subsequent effect of soil water stress on crops (phase 3). "*

The last sentence has been changed to " *This study proposes a CDI that combines three indices:* ", as suggested by reviewer 1.

**Some text pieces reflect simple negligence. Examples "representative value for loam clay according to USDA classification." Does not exist**

We apologize, this has been corrected to "clay loam".

**4.3 NDVIA insurance data were gently supplied by**

Yes, we are sorry, "gently" was a slip, it should be "kindly" " The insurance data were kindly supplied by Agroseguro."

---

## Author Comment (AC2) · 15 Jul 2019

**Referee 2**

**This paper deals with the topic of defining a new combined drought indicator (CDI) capable to anticipate crop drought events. To do so, authors combined a meteorological indicator (SPI), a soil moisture indicator (SMAI) and a vegetation indicator (NDVIA). Authors established four levels of alerts with the corresponding actions and assessed this new indicator comparing monthly alerts with crop damage provided by the agri- cultural insurance. The research carried out in this paper is of interest, and I think it is adequate to NHESS journal. The manuscript is in general well-structured and the results that follows seems very reasonable to me. Correlation between the proposed CDI and crop damage is correctly presented. It seems to me that the manuscript could be published as long as the authors answer the following comments:**

We greatly appreciate the positive evaluation of our study.

**Specific comments: 1. Authors are using a different definition of the levels of dam- age crop in the abstract and in the results or conclusions. Are the levels "watch, alert, warning type I and II" (see abstract) or "watch, warning to alert (type I and II) (see conclusions)?. Regarding Table 3 it seems to be "watch, warning, alert type I and alert type II".** In effect, there was a mistake in the abstract. Table 3 and rest of the text is the correct version, with watch, warning, alert type I and II.

**2. Could the authors extend the definition of SPI in "Methods"?. Some expla- nation of how SPI is calculated should be included to improve general understanding.**

This has been included: " *SPI is calculated by fitting the precipitation data to a gamma distribution, after which it is transformed into a normal distribution. The SPI values can then be interpreted as being the number of standard deviations by which the anomaly observed deviates from the long-term mean.* "

**3. Could the authors explain how SMAI is calculated in the studied areas?. Did the authors obtain in-situ measurements?. How did you obtain the temporal evolution of SMAI in the studied areas?**

This was done purely through modelling of the soil moisture in the soil profile. We explained this on page 5, in section 2.3, but it appears that our explanation was not clear enough. We have rewritten this part so that it is clearer to the reader. " *The deviation of the soil moisture from its long-term mean was expressed as a Soil Moisture Anomaly Index (SMAI). SMAI values were calculated for each of the five selected agricultural regions, similar to the SPI. To obtain this index, we first calculated soil moisture dynamics through the simple water balance model of Brocca et al. (2008). The long-term mean soil moisture was taken as the 10-year mean in the study period (2003-2013)....* " A full description of this water balance model and how it was parameterized is given on the next lines.

**4. Regarding your sentence: "Figure 3 shows the variation of SMAI over the studied period and for each of the five studied agricultural regions. The main two dry periods of 2004-2005 and 2011-2012 are not consistently apparent." Do the authors think that the information given by the calculated SMAI increase the accuracy of the drought prediction?.**

We believe, as stated in the cited sentence, that the impact of SMAI is not as clear as that of rainfall and vegetation stress, expressed through SMAI. Its effect appears to be clear for some pixels, but not for all of them consistently. This is in contrast with the hypothesis that precipitation deficit leads to a soil moisture deficit which, in turn, leads to vegetation stress. The reason for this is that soil moisture response to droughts is highly non-linear making prediction difficult. In other words, since the soil acts as a buffering reservoir, it complicates the response of the prediction model, and sometimes a precipitation deficit does not lead directly to a lack of soil moisture

However, as we state in the introduction, we believe it is critical to consider more than just precipitation for drought prediction. We think it is important to include soil in drought models, even though prediction becomes more complex. Another drawback is that our capacity to model soil moisture is limited on these regional scales. Future studies could focus on the use of soil moisture sensors to improve predictions.

**5. NDVIA in four pixels have been calculated for every region. Could authors explain how these pixels have been combined to obtain the NDVIA per region?. Is simply the average of the four NDVIA values?**

Yes, the average was taken. We added to the end of section 2.4. "*For each of the five regions, the final NDVIA index was then calculated based on the average of the four points or pixels of that region*."

**6. The proposed CDI seems to be a modification of Sepulcro 2012 indicator. I think some comparison with the latter, at least some advantages and drawbacks, should be included in the discussion. Is CDI the name of a family of combined indicators or is specifically the name of one indicator?. Perhaps, to avoid misunderstandings, the name of the new proposed CDI should be modified to distinguish it from the Sepulcro's CDI.**

Our indicator is indeed a modification of the Sepulcre-Cantó 2012 indicator, designed to be able to work at a finer resolution. See the discussion on this in the introduction, page 1 lines 19-20 "For the management of local policy and mitigation actions, such as farm-scale insurance schemes, smaller spatial scales than those used by Sepulcre-Cantó et al. (2012) are required."

We had already included this comparison between Sepulcre-Cantó's CDI and our new CDI in the discussion : from p.10 line 30 till page 11, line 4. We also discuss other indicators in this section.

With respect to changing the name, we strongly believe CDI to be adequate. While there might indeed be some initial confusion, we think that once the reader becomes absorbed in the text and methodologies, it is obvious where the

differences lie. We do not believe combined drought indicator" should  be a trademark name, but it could refer to any index using different (Sub)indices. Or, if you like, one could interpret our indicator as being similar to the Sepulcre-Cantó one, but simply differing in the way some variables are calculated although basically taking  into account the same 3 variables: precipitation deficit, soil moisture deficit and plant stress.

**Technical comments:**

We thank referee 2 for these technical comments, they have greatly helped to improve the text and all have been taken into account

**1. Pag. 1 – line 21/22: Review format references in the text. An example: (e.g. Wilhite 2000).**
corrected
**2. Pag. 2 – line 21: I suppose you are referring to a fig. 1 of another article. Clarify this please.**
We found it difficult to clarify and have deleted this reference.
**3. Pag. 4 – line 9: Replace "o" by "or" and "y" by "and".**
corrected
**4. Pag. 4 – line 29: What is the meaning of SPI-SL 6?**
It refers to the name of the programme code. We do not exactly know why the developers have chosen this name.
To clarify this, we have put this between " " and rephrased it as follows *" The programme "SPI_SL_6.EXE", ..."*
**5. Pag. 5 – line 2: Replace "o" by "or" and "y" by "and".**
corrected
**6. Pag. 6 – line 27: "This study proposes a CDI that combines three combines.." I suppose you want to say "three indices".**
corrected
**7. Pag. 7 – line 9: What is Agroseguro?. Explain please.**
added, " *...the provider responsible for Spanish agricultural insurance schemes.* "
More information can be found here:
*https://agroseguro.es/agroseguro/quienes-somos/introduccion-y-objetivos/introduction-and-objectives*
Agricultural insurance in Spain is based on joint participation between public and private institutions. It is voluntary, and the private insurance companies participate via a co-insurance pooling scheme. Agricultural insurance cost for producers is partly subsidized by the Government.

**8. Pag. 8 – line 5: Indicate fig. 4 is an example of the year 2004.**
added.
**9. Pag. 8 – line 23: Indicate fig. 6 shows a monthly evolution.**
added.
**10. Pag 10 – line 27 – 29: Move to Introduction. Authors should explain this Sepulcro 2012 indicator in the introduction.**

We already discussed Sepulcre-Cantó's paper in the introduction, but we have now expanded this section in order to explain it better. We believe it is appropriate to repeat the reference to their work both in the introduction and the discussion.

p.2, lines 18-22: " *The above-mentioned methods can be used to evaluate the impact of drought on agricultural productivity in regions world-wide as Sepulcre-Cantó et al. (2012) have shown for Europe. These authors proposed a combined drought indicator using SPI, fAPAR and soil moisture calculated from a regional hydrological model. For the management of local policy and mitigation actions, such as farm-scale insurance schemes, smaller spatial scales than those used by Sepulcre-Cantó et al. (2012) are required.* "

**11. Pag. 17 – Figure 3: In the first graph (3a) replace SPI-3 by SMAI.**
corrected

---

## Author Comment (AC3) · 15 Jul 2019

**Referee 3**

**General comments:**
**This paper proposes a new combined drought indicator (CDI) integrating rainfall (SPI- 3), soil moisture (SMAI) and vegetation dynamics (NDVI). It is shown that this indicator is useful to predict dry periods. Therefore the research carried out in this paper is of scientific and practical interest, and in my opinion it is adequate to NHESS journal. The manuscript is in general well structured and presented. The methodology employed and the obtained results are well exposed. It seems to me that the paper could be published as long as the authors answer the following minor concerns that arose from the review process that I made**

Thank you very much for this positive feedback. We have answered the comments above and changed the manuscript accordingly

**Specific comments:**
**I only have a couple of minor specific comments to be answered by the authors.**

**1. Did the authors made a comparison between the new combined drought indicator (CDI) that they propose and other combined drought indicators? Could the authors include in the paper some comments in this direction?**
We have included some comments in this direction in the discussion section. Please see from p.10 line 30 till page 11, line 4.

**2. Did the authors apply this new combined drought indicator on geographic areas of different characteristics with respect to the characteristics of the areas in Southern Spain analyzed in the present study? Would it be possible that the evaluation of the CDI indicator should be different?**
We agree this would be interesting. Up to now, we have only applied this study to the Andalusian region, which is already a very large with many contrasting climate and geographic situations. So, at present, applying it to other areas goes beyond the scope of this study. We expect and hope, however, that our CDI indicator would be similarly good however.

**Minor comments:**
We thank referee 3 for these comments, which have all been taken into account.

**Page 4, line 9: There are a couple of words not in English. Page 5, line 2: The same comment as I did before.**
corrected
**Page 11, line 12: The classification of CDI is not clearly exposed in this section.**
we assume that the referee refers to p.10, line 12, we have corrected the text as follows: "... are accompanied by watches, warnings and type I or II alerts of CDI in the five agricultural regions that were studied "

**Page 16, Figure 2: Perhaps it should be more clear (and homogeneous) substituting the "9" that appear at the beginning of the years, by "sep", as authors did in Figure 6. Furthermore, why the year 2013 is not printed in the figure? Also, in the description of this figure, I suppose that "La Campiña" should appear with capital letters, as in the rest of the document.**
everything corrected except for the year 2013 we were not given the data for that year by Agroseguro. We added this explanation in the material and methods, section 2.6 " Note that data for the last year of the study 2012/13 were not provided. "
**Page 17, Figure 3: Same comments that I made for Figure 2.**
corrected
**Page 19, Figure 5: Same comments that I made for Figure 2.**
corrected
**Page 19, in the bottom: I suppose that the meaning of the colors that appear at the bottom of this page should appear inside Figure 6 of page 20.**
corrected
**Page 20, Figure 6: In my opinion, this figure needs to be clearer with respect to the CDI indicator, including the meaning of the colors, for a better reader comprehension. Furthermore, it is not clear to me why the value of the agricultural crop damage intensity (blue line) is not printed for the last year.**
As mentioned a few lines earlier, we were not given this data by Agroseguro. We included this explanation in the material and methods, section 2.6 " Note that data for the last year of the study 2012/13 were not provided. "
With respect to the colours, we are not clear as to what the problem is. We have changed the scale to appear in the same figure so the reader can clearly see what every colour means. We have also, as suggested by referee 1, added that "figure 6 shows the monthly evolution of CDI", which we believe helps to now interpret this figure with more clarity. **Pages 20-21, Table 3: Could the authors improve the quality of this table?**
We have completely changed the layout of this table in order to satisfy the reviewer's request.

---

## Author Response (AR2)

Dear editor,

We thank you and the reviewers for this final review.

We have changed all the technical corrections that remained, as requested by the editor.
-We have changed the resolution of figure 5 to match that of figure 3. We additionally changed that of figure 2 to match figure 3 as well (although this was not requested).
We have original TIFF files of all figures if needed.
-We changed the scales of figure 4 from 0-1 in both a and b
-We eliminated all double spaces
-We replaced commas by digital points
However, we were not able to change the gray color of figure 1. The reason for this is that we had a data loss issue and did not have access to the original file. As the editor wrote "if possible", we hope you can understand this.

Thank you for all the assistance.
On behalf of the authors,

María Pilar Jiménez Donaire
PhD student
Department of Agronomy
University of Córdoba